# Molecular Diagnosis, Clinical Trial Representation, and Precision Medicine in Minority Patients with Oncogene-Driven Lung Cancer

**DOI:** 10.3390/cancers17121950

**Published:** 2025-06-11

**Authors:** Ahan Bhatt, Nang Yone, Mumtu Lalla, Hyein Jeon, Haiying Cheng

**Affiliations:** 1Department of Internal Medicine, NYC Health + Hospitals/Jacobi Medical Center, Bronx, New York, NY 10461, USA; bhatta4@nychhc.org; 2Department of Oncology, Albert Einstein College of Medicine, Bronx, New York, NY 10461, USA; nang.yone@einsteinmed.edu (N.Y.); mlalla@montefiore.org (M.L.); hyjeon@montefiore.org (H.J.); 3Department of Oncology, Montefiore Medical Center, Bronx, New York, NY 10461, USA; 4Department of Medical Oncology, Montefiore Einstein Comprehensive Cancer Center, Bronx, New York, NY 10461, USA

**Keywords:** NSCLC, underrepresented minorities, precision medicine, lung cancer, healthcare disparities

## Abstract

Lung cancer remains the leading cause of cancer-related mortality in the United States and worldwide. Advances in molecular profiling and targeted therapies have significantly transformed the treatment landscape for non-small cell lung cancer, especially among patients with oncogene-driven subtypes, leading to marked improvements in clinical outcomes. Despite these strides, disparities in access to molecular diagnostics and precision therapies persist, disproportionately impacting minority populations. These inequities result in lower rates of molecular testing, reduced access to specialized treatments, and underrepresentation in clinical trials, thereby limiting the generalizability of trial findings to diverse populations. In this review, we examine the current state of molecular diagnostics and precision medicine for minority patients with oncogene-driven lung cancer, emphasizing existing challenges, emerging opportunities, and future strategies to promote equity in precision oncology.

## 1. Introduction

Lung cancer is the leading cause of cancer-related deaths in the United States and globally [1], with non-small cell lung cancer (NSCLC) accounting for the majority of cases. Over the past decade, the advent of molecular profiling and targeted therapies has transformed the management of NSCLC, particularly for patients with oncologic driver alterations, such as *EGFR* mutations, *ALK* rearrangement, *RET* rearrangement, *ROS1* rearrangement, *MET exon 14* skipping alterations, *BRAF V600E*, *KRAS G12C*, and *ERRB2* (*HER2*) mutations, *NTRK1/2/3* rearrangements, and *NRG1* fusions. The development of targeted therapies for these molecular alterations has led to significant improvements in clinical outcomes, offering more effective and personalized treatment options for patients [2].

As impactful as these developments have been for patients with NSCLC, their generalizability to racial and ethnic minorities has been limited by the under-representation of minority patients in landmark clinical trials involving targeted therapies [3,4,5]. Recent studies have shown that Black and Hispanic/Latino patients have less than 5% enrollment in clinical trials, respectively, resulting in a gap in knowledge of clinical outcomes for these patients [6]. Additionally, evidence suggests that African American patients may have a distinct prevalence of driver alterations (Table 1 and Figure 1) and varying responses to treatment compared to White and Asian patients [7,8,9,10,11].

Socioeconomic determinants such as income status and access to health insurance may also constitute potential barriers to care for these populations that ultimately impact their adherence to treatments and clinical outcomes. These socioeconomic disparities have become more salient as targeted therapies have become more widely employed in the frontline setting [8,12].

In this review, we examine the current landscape of molecular diagnosis and treatment for minority patients with oncogene-driven lung cancer. We will highlight the current challenges and barriers to accessing guideline-directed standard of care as well as the differences in the prevalence of oncogenic driver mutations across populations, emphasizing the need for greater diversity in clinical research to make results more applicable to minority populations. Addressing these gaps is critical to improving survival outcomes in minority populations and ensuring equitable access to personalized lung cancer care for all patients.

**Table 1 cancers-17-01950-t001:** Frequency of genomic alterations in minority patients.

Genomic Alteration	White or Caucasian	Black or African American	Hispanic or Latino	Asian
* **EGFR ** * **mutations:** * **Exon 19** * ** Deletion** * **Exon 21 L858R** *	10–20% [13,14,15,16]	5–17% [13,14,15,16,17]	22–26% [18,19]	40–55% [13,14,15,20]
* **EGFR ** * **mutations:** * **Exon 20 ** * **Insertion**	1–2% [21]	1–2% [22]	2–3% [21,22,23]	1–4% [21,22,24]
* **KRAS ** * **mutations**	26–33% [25,26]	15–27% [13,26]	9–19% [18,23]	11–12% [25,26]
*KRAS G12C* mutation	13–15.5% [26,27]	10–11% [26,28]	7–8% [18]	3–4% [26,27]
*ALK* rearrangement	6–7% [25,29]	1–4% [26,30,31]	5–6% [12,18]	5–6% [25,32]
*ROS1* rearrangement	1–2% [26,33,34]	2–3% [26]	2% [18]	2–4% [35,36]
*MET Exon 14* Skipping Alteration	2–5% [26,37]	3–4% [26,37]	3% [18,26]	1–9% [37,38]
*RET* rearrangement	1–3% [26,39]	1–2% [26]	2% [18]	1–3% [26,40]
*ERBB2* mutation	1–2% [25,26]	<1% [16,26]	3% [18]	2–3% [26]
*BRAF V600E* mutation	3–5% [41,42]	<1% [16]	2% [18]	1–2% [43]
*NTRK* rearrangement	<1% [26]	<1% [26]	1% [18,26]	<1% [26]
*NRG1 fusion*	0–1% [44]	NR *	NR *	0–1% [44]

* Not Reported.

## 2. Molecular Diagnosis in Minority Patients: Access to Screening, NGS Testing, and Treatment

Lung cancer screening (LCS) is currently recommended by the United States Preventive Services Task Force for patients aged 50–80 years with a history of smoking 20-pack years or more and who are currently smoking or quit within the past 15 years [45,46]. This recommendation is based on the mortality benefit associated with early detection of lung cancer in 2 large, randomized trials: the National Lung Screening Trial (NLST) and the NELSON trial [47,48]. A subgroup analysis of the NLST confirmed this mortality benefit in Black patients, though only 4.5% of enrolled patients were Black [49]. The guidelines were revised in 2021 to lower the age of screening from 55 to 50 years and decrease smoking exposure from 30 pack years to 20 pack years to accommodate minority patients, who have a lower pack year smoking history and a lower age at diagnosis than White patients [45,46,50]. This change has corresponded to an increase in LCS eligibility for minority patients, with a 109% increase seen for Black patients and an 86% increase seen for Hispanic patients [51].

Unfortunately, adherence to LCS in the real world is significantly lower than in randomized clinical trials. Lopez-Olivo et al. found a pooled LCS adherence rate of 55% in a systematic review and meta-analysis of 15 cohort studies with a total of 16, 863 individuals [52]. Current smokers and non-White patients have the lowest adherence rates [52,53]. This is reflected in the current disparity of early NSCLC diagnosis in minority patients, with Black patients 15% less likely to be diagnosed early than White patients, and Latino, Asian American, and Pacific Islander patients are 17% less likely to be diagnosed early compared to White patients [54].

Current standard practice after the confirmation of a new diagnosis of advanced NSCLC includes tissue- and/or liquid-based next-generation sequencing (NGS) to identify targetable genomic alterations and guide treatment decisions. More comprehensive molecular analysis with the identification of co-mutations, tumor mutational burden, and non-coding RNA can further optimize treatment selection [55]. Unfortunately, the prevalence of testing in the United States is only 50–60%, with a greater likelihood of testing in patients who have favorable socioeconomic strata, insurance access, a younger age, and access to hospitals and clinical trials [56]. Asians, Blacks, and Hispanics are less likely than Whites to receive the recommended molecular testing necessary for the receipt of targeted therapies [57]. This was demonstrated by Bruno et al., who illustrated that Black patients with NSCLC are less likely than White patients to undergo NGS testing before first line of therapy (36.6% versus 29.7%, *p* < 0.0001) or at any time point thereafter (54.7% versus 43.8%, *p* < 0.0001) [58].

Cost remains a significant barrier to accessing genomic and NGS testing, which is compounded by variability in insurance coverage [59]. Additionally, historical disparities in the healthcare system may have contributed to varying levels of patient trust, especially in the adoption of newer technologies [60]. Disparities in access to molecular testing can lead to suboptimal treatment outcomes and higher treatment-related mortality. This is particularly concerning for patients with certain targetable mutations (e.g., *EGFR*, *ALK*), who may face an increased risk of immune-related adverse events when treated with immunotherapy prior to starting targeted therapy due to delayed molecular testing [61].

Disparities in access to treatment also persist for minority patients. Studies suggest that Latino individuals are 30% less likely to receive treatment than White patients [54,62]. Black patients are less likely to receive surgical treatment than White patients for early-stage disease after adjustment for age, insurance status, TNM stage, income, and treatment at academic versus community hospitals [63]. These factors, combined with other barriers such as lack of awareness about clinical trials, restricted access to trial opportunities, socioeconomic constraints, competing responsibilities, provider biases, and clinical trial study design and enrollment criteria, also contribute to the underrepresentation of minority populations in clinical research. Addressing these challenges is essential to building a culturally competent healthcare structure and advancing equitable healthcare outcomes [64,65].

## 3. Treatment of Minority Patients with Oncogene Drivers: The Lack of Representation in Landmark Clinical Trials

Disparities in minority access to clinical trials have extended into the evaluation of targeted therapies for oncogene-driven NSCLC, which have played a pivotal role in reshaping the treatment paradigm of this disease. As a result, real-world population-based studies have increasingly been used as surrogates to clinical trials to help generalize findings to underrepresented populations. In the following sections, we review the common genomic oncogenic driver alterations with FDA-approved targeted therapies and assess the representation of minoritized populations in both landmark clinical trials and real-world settings. The mechanisms of action for currently approved targeted therapies are illustrated in Figure 2. Key clinical trials and real-world studies supporting these therapies are summarized in Table 2 and Table 3, respectively.

### 3.1. EGFR Mutations

*EGFR* mutations are common targetable mutations in NSCLC, occurring in 10–20% of Caucasian patients and up to 40–50% of Asian patients [13,14,15,20]. These mutations are also common in Hispanic patients, occurring at a frequency of 22–26% [18,19]. In African American patients, these mutations are less common, occurring at a frequency of 5–17% [13,14,15,16,17,66]. 85–90% of all *EGFR* mutations in NSCLC consist of Exon 19 deletions and point mutations of L858R in Exon 21 [67].

Osimertinib, the third-generation *EGFR* tyrosine kinase inhibitor (TKI), is the current standard first-line TKI for *EGFR*-mutated (*exon 19* deletion or *L858R*) NSCLC. Osimertinib demonstrated superiority over earlier generations of *EGFR* TKIs (erlotinib and gefitinib) in the FLAURA trial with median progression-free survival (PFS) of 18.2 months as compared to 10.2 months with erlotinib and gefitinib (HR 0.46, CI 0.37–0.57) and a mean overall survival (OS) of 38.6 months as compared to 31.8 months with erlotinib and gefitinib (HR 0.8, 95% CI 0.64–1) [68,69]. More recently, an improved PFS has also been shown with the addition of chemotherapy to osimertinib in the FLAURA-2 study [3] and with the combination of amivantamab, an *EGFR*-*MET* bispecific antibody, and lazertinib, another third-generation *EGFR* TKI, in the MARIPOSA study [70]. There is ongoing discussion on which patients benefit from first-line treatment intensification and how to incorporate these recent studies into the standard treatment paradigm for advanced *EGFR-mutated* NSCLC.

There is, however, limited data from these pivotal trials on the outcomes for African American and Hispanic patients with advanced *EGFR*-mutated NSCLC. Most patients enrolled in these trials were White and Asian, with ≤1% of African American patients in the FLAURA and FLAURA 2 trials [3,69] and 5% of other patients (Black or African American, American Indian or Alaska Native, multiple, unknown, and not reported) enrolled in the MARIPOSA trial [70]. The enrollment of Hispanic patients is not specified in any of the trials [3,69,70].

Real-world analysis on the outcomes of osimertinib in ethnic minorities has shown equivalent outcomes in Hispanic patients and worse outcomes in African American patients [19,23]. Inferior survival outcomes have been previously described in real-world analysis of African American patients on erlotinib and gefitinib for *EGFR-mutated* lung cancer [17]. In contrast, real-world analysis of Hispanic patients on *EGFR* TKI has shown similar outcomes to non-Hispanic patients with a median PFS of 14.4–15.9 months and OS of 32 months [19,23].

The use of osimertinib has also recently been expanded to earlier stages of NSCLC.

**Table 2 cancers-17-01950-t002:** Treatment of minority patients with oncogene drivers: the lack of representation in landmark clinical trials.

Genomic Alteration	Treatment	Key Trials (Phase, *n*)	Racial/Ethnic Breakdown (%)				OS (%)/mOS	PFS (mo)	Other Endpoints	Ref
			Black	Hispanic	Asian	White				
* **EGFR Exon 19 ** * **deletion and ** * **Exon 21 L858R** *	Osimertinib	FLAURA (III, 556)	-	-	62	36	3 years OS 28 (Osi) vs. 9 (GE)	18.9 (Osi) vs. 10.2 (GE)	ORR 80% (Osi) vs. 76% (GE)	[69]
	Osi ± C	FLAURA2 (III, 557)	<1	-	64	28	2 years OS 79 (Osi + C) vs. 73 (Osi)	25.5 (Osi + C) vs. 16.7 (Osi)	ORR 92% (Osi + C) vs. 83% (Osi)	[3]
	Osimertinib	ADAURA (III, 682)	-	-	64	-	5 years OS 88 (Osi) vs. 78 (P)	NE	DFS 65.8 (Osi) vs. 28.1 (P)	[71,72,73]
	Osimertinib	LAURA (III, 216)	-	-	81	-	NR (interim)	39.1 (Osi) vs. 5.6 (placebo)	ORR 57% (Osi) vs. 33% (P) DoR 36.9 mo (Osi) vs. 6.5 mo (P)	[74]
	Ami + Laz	MARIPOSA (III, 858)	<1	-	58	38	2 years OS 74 (Ami + Laz) vs. 69 (Osi)	23.7 (Ami + Laz) vs. 16.6 (Osi)	ORR 86% (Ami + Laz) vs. 85% (Osi)	[70]
	Ami + C ± Laz	MARIPOSA-2 (III, 657)	-	-	48	48	NR (interim)	6.3 (Ami + C) vs. 8.3 (Ami + Laz + C) vs. 4.2 (C)	ORR 64% (Ami + C) vs. 63% (Ami + Laz + C) vs. 36% (C)	[5]
* **EGFR Exon 20 ** * **insertion**	Amivantamab	CRYSALIS (I, 158)	<1	-	56	35	22.8 mo (Ami)	8.3 (Ami)	ORR 40%, DOR 11.1 mo	[75]
		PAPILLON (III, 308)	<0.01	-	60	35	2 years OS 72 (Ami + C) vs. 54 (C)	11.4 mo (Ami + C) vs. 6.7 (C)	ORR 73% (Ami + C) vs. 47% (C)	[76]
* **KRAS G12C ** * **mutation**	Sotorosib	CodeBreaK 100 (I/II, 126)	2	-	15	82	12.5 mo	6.3	ORR 41%	[77]
		CodeBreaK 200 (III, 345)	<0.01	-	12	83	10.6 mo (Soto) vs. 11.3 mo (Dt)	5.6 (Soto) vs. 4.5 (Dt)	ORR 28.1% (Soto) vs. 13.2% (Dt)	[78]
	Adagrasib	KRYSTAL-1 (I/II, 116)	8	-	4	84	12.6 mo (interim)	6.5	ORR 42.9%	[79]
* **ALK** * **rearrangement**	Alectinib	ALEX (III, 303)	-	-	46	-	NR (interim for both groups)	34.8 (Alec) vs. 10.9 (Cri)	ORR 82.9% (Alec) vs. 75.5% (Cri)	[80]
		ALINA (III, 257)	<0.01	-	56	42	NR (interim)	DFS NR (Alec) vs. 41.3 mo (C)	3y-DFS 88.7% (Alec) vs. 54% (C)	[4]
	Brigatinib	ALTA-1L (III, 275)	-	-	39	-	4 years OS 66 (Bri) vs. 60 (Cri)	24 (Bri) vs. 11 (Cri)	ORR 71% (Bri) vs. 60% (Cri)	[81,82]
	Lorlatinib	CROWN (III, 296)	<0.01	-	44	49	NR (second interim)	NR (Lor) vs. 9.1 (Cri)	ORR 81% (Lor) vs. 63% (Cri)	[83,84]
* **ROS1 ** * **rearrangement**	Entrectinib	*ALK*A-372-001 (I, 1) STARTRK-1 (I, 2) STARTRK-2 (II, 51)	-	-	13	80	21 mo	11.2	ORR 57%, DOR 10.4 mo	[85]
	Repotrectinib	TRIDENT-1 (I/II, 127)	-	-	59	34	NE (TN), 25.1 mo (PT)	35.7 (TN), 9 (PT)	ORR 79% (TN), 38% (PT) DOR 34.1 mo (TN), 14.8 mo (PT)	[86]
* **MET ** * **Exon 14 skipping mutation**	Capmatinib	GEOMETRY mono-1 (II, 160)	-	-	19	77	NR (expansion cohort)	12.4 (TN) 5.4 (PT)	ORR 68% (TN), 41% (PT) DOR 12.6 mo (TN), 9.7 mo (PT)	[87,88]
	Tepotinib	VISION (II, 99)	-	-	21	75	NR (interim)	8.5	ORR 46%, DOR 11.1 mo	[89]
* **RET ** * **rearrangement**	Selperactinib	LIBRETTO -001 (I/II, 316)	<1	-	41	49	3 years OS 66 (TN), 57 (PT)	22 (TN), 26.2 (PT)	ORR 82.6% (TN), 61.5% (PT)	[90,91]
	Pralsetinib	ARROW (I/II, 233) *	-	-	39	52	NR (interim)	10.9-NR (TN) vs. 12.8–16.5 (PT)	ORR 68–79% (TN), 59–73% (PT)	[92]
* **BRAF V600E ** * **mutation**	Dabrafenib/Trametinib	NCT01336634 (II, 93)	3	-	8	85	5 years OS 22 (TN), 19 (PT)	10.8 (TN), 10.2 (PT)	ORR 63.9% (TN), 68.4% (PT)	[93]
	Encorafenib/Binimetinib	PHAROS (II, 98)	3	-	7	88	NE	NE (TN), 9.3 (PT)	ORR 75% (TN), 46% (PT)	[94]
* **ERBB2 ** * **mutation**	T-Dxd	DESTINY-Lung 01 (II, 91)	1	-	34	44	17.8 mo	8.2	ORR 55%; DOR 9.3 mo	[95]

Ada, adagrasib; Alec, alectinib; Ami, amivantamab; Bri, brigatinib; C, platinum-doublet chemotherapy; Cri, crizotinib; DFS, disease free survival; DOR, duration of response; Dt, docetaxel; GE, gefitinib or erlotinib; HR, hazard ratio; Laz, lazertinib; Lor, lorlatinib; Mo, months; NE, Not evaluated; NR, not reached; ORR, objective response rate; OS, overall survival; Osi, osimertinib; P, placebo; PFS, progression-free survival; PT, previously treated; Ref, references; Soto, sotorosib; T-Dxd, trastuzumab deruxtecan; TN, treatment naiive; VS, versus. * Eligibility changes.

**Table 3 cancers-17-01950-t003:** Treatment of minority patients with oncogene drivers: clinical outcomes in real-world studies.

Mutation	Treatment	Racial/Ethnic Background	ORR	OS (95% CI)	PFS (95% CI)	Ref
* **EGFR Exon 19 ** * **Deletion and ** * **Exon 21 L858R** *	erlotinib, gefitinib, afatinib	Black 36% Non-Black 64%	Blacks 63.6% Non-Black 76.0%	2-year survival rates: Black 33.3% Non-Black 61.3%	NA	[17]
	*EGFR* TKIs	Hispanic 100%	60.5% (52.10–69.09)	32 mo (12.4–20.6 mo)	15.9 mo (12.4–20.6 mo)	[23]
	Osimertinib	Hispanic 100%	NA	NA	14.4 mo (95% CI 12.4–18.2 mo)	[19]
* **EGFR ** * * **Exon ** * * **20 ** * **Insertion**	C 54% IO 15% EGFR Targeted Therapy 12% Surgery 24%	White 63% Asian 3.1% Black 9% Hispanic 2%	NA	23.8 mo in the entire group 17 mo in Stage IV patients	NA	[22]
	C 52.7% C + Bev 11.8% C+ Pem 4.3% 1G TKIs 21.5% Osi 3.2% Afatinib 3.2% *Exon20* inh 5.2%	Hispanic 100%	NA	C + Bev 18 vs. 14.5 mo (HR = 0.57, 0.36–0.90) 1 G TKIs 16.36 vs. C 14.5 mo (HR = 0.49, 0.25–0.98) Osi 19.4 vs. C 14.5 mo (HR = 0.13; 0.02–0.82) *Exon20* inhibitors 25.6 vs. C 15 mo (HR = 0.26, 0.06–1.07)	C + Bev 5.35 vs. C 4.8 mo (HR = 0.47; 0.24–0.89) TKIs 6.6 vs. C 4.8 mo (HR = 0.36, 0.21–0.61) C + Pem 5.53 vs. C 4.8 mo (HR = 0.25, 0.06–1.01)	[96]
* **KRAS G12C ** * **mutation**	sotorasib v docetaxel	White 71% Black 11%	NA	≥2L Setting sotorasib 10.2 mo (8–14.6 mo) docetaxel 7.2 mo (5.1–10.6 mo)	NA	[97]
	sotorasib	White 77% Black 19% Hispanic 2%	34%	12 mo (10.2–16.5)	6 mo (4.3–8.1)	[98]
	sotorasib	White 81% Asian 4% Black 6% Hispanic 3%	28%	12.6 months (8.3 mo—NA)	5.3 months (3.6–6.6 mo)	[99]
* **ALK ** * **Rearrangements**	alectinib 59.8% crizotinib 40.2%	White 45.3% Asian 30.8% Black 2.6%	NA	alectinib 54.1 mo crizotinib 45.8 mo	alectinib 29.3 mo crizotinib 10.4 mo	[100]
	alectinib 30.4% crizotinib 69.6%	alectinib White 60.3% Asian 14.9% Black 4.3% Hispanic 3.5% crizotinib White 65.9% Asian 5.4% Black 6.3% Hispanic 5.8%	alectinib 78.7% (71–85.2) crizotinib 48.9% (42.4–55.6)	alectinib NR (29.2-NR) crizotinib 23 mo (17–33.5)	alectinib 24.5 mo (15.8-NR) crizotinib 12 mo (9.3–14.4)	[101]
* **ROS1 ** * **Rearrangements**	entrectinib, crizotinib	entrectinib White 49% Asian 44% Black 4% Hispanic 1% crizotinib White 54% Asian 9% Black 12% Hispanic 17%	NA	entrectinib NR crizotinib 18.5 mo (15.1–47.2 mo)	entrectinib 16.8 mo (12–26.3 mo) crizotinib 8.2 mo (6.5–9.9 mo)	[102]
* **MET ** * * **Exon ** * * **14 ** * **Skipping Mutation**	capmatinib 50.9% IO 14.3% C 14.6% C + IO 15%	White 49.1% Black 26.8% Hispanic 9.4%	capmatinib 73.4% IO 68.6% C 52% C + Io 54.8%	capmatinib NE IO NE (14.3 mo-NE) C 17.6 mo (10.9 mo-NE) C + IO 29.9 months (20.2–32.1 mo)	capmatinib NE IO 12.6 mo (11.1 mo-NE) C 10.1 mo (5.9 mo-NE) C + IO 12 months (9–12.6 mo)	[103]
	capmatinib	White 79.4% Asian 7.4% Black 13.2% Hispanic 19.1%	(1 + 2L) 85.3% (74.6–92.7) 1L 90.9% (80.1–97.0) 2L 61.5% (31.6–86.1)	1L 14.1 mo (13.9 mo-NE)	Any line 14.5 mo (14.1 mo-NE) 1L 14.1 mo (10.1 mo-NE)	[104]
* **RET ** * **Rearrangements**	selpercatinib	Asian 10% Non-Asian 90%	Any line 68% 1L 69% ≥2L 68%	NA	Any line 15.6 mo (8.8–22.4 mo) 1L 15.6 mo ≥ 2L 12.2 mo	[105]

1G, first generation; 1L, first line; 2L, second line; Bev, Bevacizumab; C, chemotherapy; *Exon20* inh anti-*EFGR Exon20* inhibitor agents (Amivantamab, Mobocertinib); HR, Hazard Ratio; IO, immunotherapy; Mo, months; NA, Not available; NE, not evaluable NR, not reached; ORR, objective response rate; OS, overall survival; Osi, osimertinib; PFS, progression-free survival; Ref: References; Pem, pembrolizumab; TKI, tyrosine kinase inhibitor.

The ADAURA trial showed an 80% reduction in the risk of disease recurrence or death with three years of adjuvant osimertinib (HR 0.27, 0.21–0.34) [71,72]. Benefit of adjuvant osimertinib after chemoradiation has also been shown for patients with stage III NSCLC in the LAURA trial, which showed a median PFS of 39.1 months with osimertinib versus 5.6 months with placebo (HR 0.16, 0.10–0.24) [74]. Similarly to the trials for advanced *EGFR-*mutated NSCLC, most patients enrolled in the ADAURA and LAURA trials were also Asian [71,74].

*EGFR* Exon 20 insertion mutations are the third most common *EGFR* mutations, accounting for 9% of all *EGFR* mutations [21]. They are similarly more common in Asian patients (1–4%), with a lower prevalence reported in Hispanic (2–3%) and African American patients (1–2%) [21,22,23,24,106]. Amivantamab was approved as a second-line treatment after progression on platinum doublet chemotherapy (PDC) based on the phase 1 CHRYSALIS trial, with an objective response rate (ORR) of 40% with median PFS of 8.3 months, and a median OS of 22.8 months [75]. More recently, the combination of amivantamab and PDC has been approved as a first-line treatment for locally advanced or metastatic NSCLC with *EGFR exon 20* insertions: The PAPILLON trial reported a median PFS of 11.4 months with the combination of amivantamab-PDC versus 6.7 months with PDC alone (HR 0.40, CI 0.3–0.53) [76].

Real-world studies predating the use of amivantamab have shown a limited median OS of 14–16 months in stage IV patients with *EGFR* exon 20 insertion mutations [22,96]. Interestingly, a review of the ASCO CancerLinQ Discovery data set did show significantly lower risk of death in African American patients as compared to White patients (HR 0.62, 0.42–0.91), but the stage and treatment for these patients were not specified [22]. A more recent real-world analysis of a cohort of Hispanic patients with NSCLC included 6 patients who received amivantamab and mobocertinib had a median OS of 25.6 months versus 15 months with chemotherapy (HR 0.26, 0.06–1.07), showing efficacy of *EGFR Exon 20* insertion inhibiting agents in this population [96].

### 3.2. KRAS Mutations

*KRAS* mutations are among the most prevalent driver oncogenes in patients with NSCLC, occurring in 25–30% of non-squamous NSCLC [107]. Unlike *EGFR* mutations, *KRAS* mutations are generally associated with smoking and are more commonly found in Caucasians (26–33%) and African Americans (15–27%) than Asians (11–12%) [13,25,26]. Among *KRAS* mutations, *G12C* is the most common mutation and is similarly more common in Caucasians (13–15.5%) than African Americans (10–11%) and Asians (3–4%) [26,27]. The frequency of *KRAS* mutations in Hispanic patients is 9–19%, with *G12C* similarly being the most frequent mutation, with a 7–8% prevalence [18,23].

Unlike *EGFR*-mutated or *ALK*-positive cases, NSCLC patients with *KRAS* mutations have shown favorable responses to combination chemoimmunotherapy in the first-line setting [108]. Unfortunately, patients who progress beyond the first line of treatment have a poor prognosis with second-line docetaxel with or without antiangiogenic therapy or single-agent pemetrexed [109,110,111]. Sotorasib and adagrasib are the two *KRAS G12C* inhibitors that were approved for clinical use in the second-line setting and beyond based on their efficacy, safety, and tolerability demonstrated in the phase 2 CodeBreaK 100 and KRYSTAL-1 trials, respectively [77,79,112]. Phase 3 trials comparing the use of sotorasib and adagrasib to docetaxel have confirmed meaningful improvements in median PFS (5.6 months sotorasib versus 4.5 months docetaxel; HR 0.66 in CodeBreaK 200 and 5.49 months adagrasib versus 3.84 months docetaxel; HR 0.58 in KRYSTAL-12), but have not detected a significant difference in OS (median OS of 10.6 months with sotorasib versus 11.3 months with docetaxel; HR 0.96) [78,113]. Most patients enrolled in CodeBreaK 200 for sotorasib were Caucasian; 1.2% of enrolled patients were African Americans [77,79,112]. A greater proportion of African Americans (7.8%) were enrolled in KRYSTAL-1 [78,79,112,113].

Real-world studies (which have included 6–19% African Americans) have confirmed the effectiveness of sotorasib, seen in clinical trials, with a median PFS of 5–6 months and median OS of 10–12 months [97,98,99]. No difference has been seen in response and survival rates according to race. Even though sotorasib was not seen to provide an OS benefit in clinical trials, it remains an option for patients who are unable to tolerate chemotherapy. OS with docetaxel in clinical trials and real-world studies has also been discordant, with docetaxel demonstrating a longer overall survival benefit in clinical trials than in the real-world setting [97].

### 3.3. ALK Rearrangements

Rearrangements of the *ALK* gene are present in 3–5% of NSCLC [4,114]. Patients with *ALK* rearrangements are typically younger, non-smokers, and have a higher likelihood of developing brain metastasis [80]. The frequency of African American patients with *ALK* rearrangements is lower (1–4%) than that seen in White, Asian, and Hispanic patients [26,31].

Alectinib, brigatinib, and lorlatinib are three next-generation *ALK* inhibitors that are currently FDA-approved for the treatment of *ALK*-positive advanced NSCLC. Alectinib, a second-generation *ALK* inhibitor, was evaluated against crizotinib in the ALEX study and showed a median PFS of 35 months (versus 11 months with crizotinib; HR 0.43), lower time to central nervous system (CNS) progression (12% with alectinib versus 45% with crizotinib; HR 0.16) and lower frequency of grade 3–5 adverse events [80,114]. Brigatinib is another second-generation *ALK* inhibitor that showed comparable efficacy to alectinib against crizotinib in the ALTA 1L trial, though it has been associated with more toxicities, including increased creatinine kinase level and interstitial lung disease/pneumonitis [81,82]. Lorlatinib, a third-generation *ALK* inhibitor, was evaluated against crizotinib in the CROWN trial with an ORR of 76% (versus 58% with crizotinib) and an impressive 5-year PFS of 60% (versus 8% with crizotinib). Unfortunately, the increased efficacy observed with lorlatinib does come at the cost of more side effects, including hyperlipidemia, edema, and neurocognitive events [83,84,115].

A majority of Caucasian and Asian patients were recruited in the ALEX, ALTA 1L, and CROWN trials [80,82,83]. Real-world studies (including 2–4% African American patients and 3–5% Hispanic patients) have confirmed the effectiveness of alectinib against crizotinib, with a median PFS of 25–29 months (versus 10–12 months with crizotinib) [100,101]. Limited real-world studies are available for brigatinib and lorlatinib use in minority patients.

The use of *ALK* inhibitors, like *EGFR* inhibitors, has also been expanded to earlier stages of NSCLC with the ALINA trial that showed a 2-year disease-free survival of 93.8% for patients with stage II or IIIA *ALK*-positive NSCLC with adjuvant alectinib (versus 63% with PDC; HR 0.24) [4]. Adjuvant alectinib was also associated with an improvement in CNS disease-free survival (HR 0.22, CI 0.08–0.58). Unlike the ADAURA trial in early-stage *EGFR*-mutated NSCLC, patients did not receive adjuvant PDC in addition to alectinib in the ALINA trial. African American patients were also under-represented in the trial (<1% of patients), limiting its generalizability [4].

### 3.4. Other Targetable Mutations

*ROS1* rearrangements, *MET Exon 14* skipping mutations, *RET* rearrangements, *ERBB2* mutations, *BRAFV600E* mutations, *NTRK* rearrangements, and *NRG1* fusions are all rare driver mutations in NSCLC. It is therefore more difficult to determine the characteristics and outcomes of populations with these mutations, to recruit sufficient patients for large phase 3 trials to evaluate the effectiveness of targeted treatments, and to analyze the use of targeted medications in broader populations in real-world studies.

***ROS1*** rearrangements occur in 2% of patients with NSCLC and are commonly associated with brain metastasis [85,86,102,116]. Entrectinib, a *ROS1*, *TRK*, and *ALK* TKI, has become a key treatment option for patients with *ROS1*-rearranged NSCLC. Its approval was supported by three phase 1/2 studies (*ALK*A-372-001, STARTRK-1, and STARTRK-2), which demonstrated deep and durable responses across a broad range of solid tumors with *ROS1* rearrangements [85]. In *ROS1* fusion-positive NSCLC, entrectinib demonstrated an ORR of 68%, a median PFS of 15.7 months, and a median OS of 47.8 months [85,102,116]. Entrectinib also has better CNS penetration than its predecessor, crizotinib, with an intracranial ORR of 80%, although response is more limited in patients with prior crizotinib exposure.

Repotrectinib, a *ROS1* and *TRK* TKI, was recently approved after the phase 1/2 TRIDENT-1 study showed an ORR of 79%, 89% intracranial response rate, and median PFS of 35.7 months in patients with no prior exposure to TKIs [86]. Patients who had prior exposure to TKIs (crizotinib 82%, entrectinib 16%) had an ORR of 21% with a median PFS of 9 months and median OS of 25.1 months. Limited real-world analysis is present to confirm the efficacy of entrectinib and repotrectinib.

***MET*** ***Exon 14*** skipping mutations occur in 3–4% of patients with NSCLC, with equivalent distribution among race and geographic region [18,26,37]. Capmatinib and tepotinib are two selective *MET* inhibitors that were approved for use in newly diagnosed and relapsed patients based on their efficacy and tolerability in the phase 2 GEOMETRY mono-1 and VISION studies, respectively [87,88,89,104]. In the first-line setting, response rates are 57.3% with tepotinib, with a median PFS of 12.6 months and median OS of 21.3 months. In the second line setting, response rates are 45% with tepotinib, with a median PFS of 11 months and median OS of 19.3 months [89,104]. Similar efficacy data are observed with capmatinib.

Both phase 2 trials evaluating selective *MET* inhibitors enrolled mostly Caucasian (~75%) and Asian patients (~20%). Real-world analyses have confirmed the efficacy of first-line capmatinib in a more diverse population (Black 26.8% and Hispanic 9.4%) with a response rate of 73.4% (versus 52% with chemotherapy alone and 54.8% with chemotherapy and immunotherapy) [103].

***RET*** rearrangements are oncogenic drivers of 1–2% of NSCLC [90]. Similarly to *EGFR* and *ALK* alterations, patients with *RET* rearrangements are less responsive to chemoimmunotherapy in the first-line setting for advanced and metastatic disease [117]. Selpercatinib, a highly selective *RET* inhibitor, was compared to PDC with or without pembrolizumab in the phase 3 LIBRETTO-431 trial and demonstrated an ORR of 84% (versus 65% with chemoimmunotherapy), intracranial response rate of 82% (versus 58% with chemoimmunotherapy) and median PFS of 24.8 months (versus 11.2 months with chemoimmunotherapy; HR 0.46) [90,118]. Pralsetinib is another selective *RET* inhibitor that has been approved for use in patients with *RET* fusion-positive NSCLC based on an ORR of 72% seen in the phase 1/2 ARROW trial. Real-world analysis has corroborated the benefit of *RET* inhibitors with an ORR of 70% and median PFS of 15–16 months, but the populations evaluated in both clinical trials and real-world studies have been mostly Asian and Caucasian [105].

***ERBB2*** ***(HER2)*** mutations occur in ~1–3% of NSCLC [95]. Historically, *HER2*-targeted therapy has had a limited response in NSCLC, and patients have been treated with standard first-line chemoimmunotherapy, with limited options on progression [119,120]. Trastuzumab deruxtecan (T-Dxd) is an antibody-drug conjugate that consists of a humanized anti-*HER2* monoclonal antibody linked to a topoisomerase I inhibitor payload that has received tumor-agnostic approval for *HER2*-positive solid tumors on immunohistochemistry. The use of T-Dxd in the second or greater line of treatment for patients with *HER2*-mutated NSCLC was evaluated in the DESTINY-Lung01 phase 2 study and showed an ORR of 55% with a median PFS of 8.2 months and median OS of 17.8 months [95].

Activating mutations in ***BRAF*** occur in 3–5% of patients with NSCLC [41,42,43,121,122]. The majority of *BRAF* mutations (≥50%) occur on codon 600 (*V600E*). *BRAF V600E* mutated NSCLC has been associated with varying responses to first-line chemoimmunotherapy [42,121,123,124]. Given the efficacy and tolerability of the combination of *BRAF* and MEK inhibitors (dabrafenib and trametinib, encorafenib and binimetinib) in *BRAF V600E*-mutated melanoma, the combination of *BRAF* and MEK inhibitors was evaluated in *BRAF V600E*-mutated NSCLC in the phase 2 NCT01336634 and PHAROS studies [125,126,127,128]. The ORR in treatment-naive patients was 65–75% with a median PFS of 30.2 months seen with the combination of encorafenib and binimetinib. In previously treated patients, response rates were 45–63% with a median PFS of 9 months and OS of 17–22 months [93,125,126,127]. Real-world studies have confirmed an ORR of 74.1%, PFS of 19.9 months, and OS of 29.9 months in treatment-naive patients, though in a predominantly Caucasian population [129].

***NTRK*** **1/2/3** rearrangements are extremely rare oncogenic drivers in NSCLC, occurring in ~0.2% of cases [85,116]. Larotrectinib and entrectinib both have tumor-agnostic approval in the treatment of *TRK* fusion-positive cancers and have demonstrated an ORR of 60–65% in NSCLC with durable responses (median PFS of 22 months and median OS of 39.3 months with larotrectinib) [85,86,116,130].

***NRG*****1** fusions are similarly rare oncogenic drivers of NSCLC, occurring in ~0.2–0.5% of cases [131]. Their presence is enriched in invasive mucinous adenocarcinomas, where incidence has been documented to be as high as 30% in Asian and Caucasian patients [131]. Zenocutuzumab, an IgG1 bispecific antibody directed against *HER2* and *HER3*, was recently approved for patients with advanced *NRG1* fusion-positive NSCLC and pancreatic cancer after it demonstrated an ORR of 29% and a median duration of response of 12.7 months in NSCLC in the eNRGy trial [132].

## 4. Impact of Racial and Socioeconomic Disparities on NSCLC Outcomes

Advances in lung cancer screening, molecular diagnostic techniques, and the advent of targeted treatment have improved mortality for patients with NSCLC. However, non-Hispanic Black men continue to be disproportionately affected with the highest incidence and mortality from lung cancer compared to other racial and ethnic groups [133,134]. Black patients are more likely to be diagnosed with lung cancer at a younger age and to have more advanced-stage disease at diagnosis compared to their White counterparts [64,134]. As highlighted by this review, evidence suggests that Black patients also have a distinct prevalence of driver alterations and worse outcomes on certain targeted therapies. Understanding the biological and socioeconomic factors driving disparities in treatment outcomes is critical in optimizing clinical responses.

Distinctions in risk factors for NSCLC, such as smoking and smoking-related comorbidities, contribute to the disparity in lung cancer incidence and outcomes in racial and ethnic minorities [64,133,135]. Risk of lung cancer correlates more with smoking duration than intensity and hence is higher in Black patients, who tend to smoke fewer packs per day but for a greater number of years [64,136,137]. Differential rates of tobacco cessation after diagnosis can also be seen with increased nicotine dependence from the use of mentholated cigarettes and weaker social supports [64,133,135]. Smoking additionally drives neoantigen richness and somatic mutations, causing heterogeneity across the tumor microenvironment, and may explain the differences in oncogenic driver mutation prevalence and limited responses seen with targeted therapies in minoritized populations.

Racial and ethnic disparities in social determinants of health (i.e., insurance status, transportation, access to primary care) can further exacerbate adverse outcomes [64,133,135]. Geographic and economic disparities in minoritized communities can reduce access to high-quality healthcare, including lung cancer screening, molecular diagnostic testing, and targeted treatments, and have been linked to a higher prevalence of comorbidities such as cardiovascular disease and HIV [138]. Limited job prospects and housing availability for economically compromised populations can lead to significantly increased exposure to workplace carcinogens (i.e., asbestos, silica, radon, arsenic), increasing the risk for developing NSCLC [64,136,139]. Redlining has additionally been associated with later-stage cancer diagnosis, decreased likelihood of receiving guideline-concordant treatment, and increased cancer mortality [140].

## 5. Future Directions

The next step in ensuring equitable advancements in NSCLC treatment is to improve minority representation in clinical trials. We have so far discussed many improvements and advancements in targeted treatment for patients with NSCLC with actionable gene alterations. However, representation of minorities—particularly Black and Hispanic populations—has remained low, limiting the generalizability of available data. While real-world data provides important insights into the effectiveness and safety of targeted therapies across diverse racial and ethnic populations, prospective clinical trials remain essential for confirming these outcomes and guiding treatments.

Enhanced recruitment strategies tailored to underrepresented populations are needed to address barriers to participation [141,142]. These efforts include engaging community physicians, patient advocacy organizations, and culturally competent study coordinators to raise awareness of trial opportunities. Inclusive trial designs should be developed with input from diverse patient populations to reflect broader clinical needs, such as addressing comorbidities and socioeconomic factors. Regulatory agencies and sponsors can play a vital role by incentivizing inclusive enrollment through targeted funding, public–private partnerships, and diversity reporting requirements.

To overcome barriers to recruiting and retaining underrepresented populations, logistical challenges such as transportation, financial toxicities, as well as cultural competencies and education must be addressed [143]. Additionally, implementing telehealth solutions and decentralized trial models can increase access for patients in remote or underserved areas [144]. Offering financial support, such as travel reimbursement and modest compensation, can also alleviate the economic burdens that disproportionately affect minority populations. Sustained funding from public and private sources is crucial to support these efforts. Equally important is culturally competent outreach and education, which includes providing multilingual materials and partnering with community leaders and organizations to foster trust and dispel misconceptions about clinical research [145].

Finally, translational research is needed to explore whether intrinsic genomic differences or external access-to-care issues contribute to observed disparities in the outcomes of minority patients. Large-scale databases and real-world registries that include robust demographic details can help identify trends, refine therapeutic strategies, and guide clinical decision-making. Multidisciplinary collaborations involving oncologists, community outreach, as well as research coordinators can develop comprehensive approaches to address these disparities. Together, these efforts will ensure that the benefits of precision oncology are equitably extended to all populations.

## 6. Conclusions

Molecularly targeted therapies have revolutionized the treatment landscape for NSCLC by offering personalized, mutation-driven approaches. However, the underrepresentation of Black and Hispanic patients in clinical trials remains a critical limitation that hinders our understanding of outcomes and management in these populations. Real-world evidence suggests that while Hispanic patients often achieve comparable results to non-Hispanic patients, Black patients may experience worse outcomes on certain targeted therapies, underlining the urgent need for more robust prospective data. Overcoming socioeconomic and logistical barriers to trial participation, ensuring inclusive study designs, and dedicating sufficient financial resources are essential next steps.

## Figures and Tables

**Figure 1 cancers-17-01950-f001:**
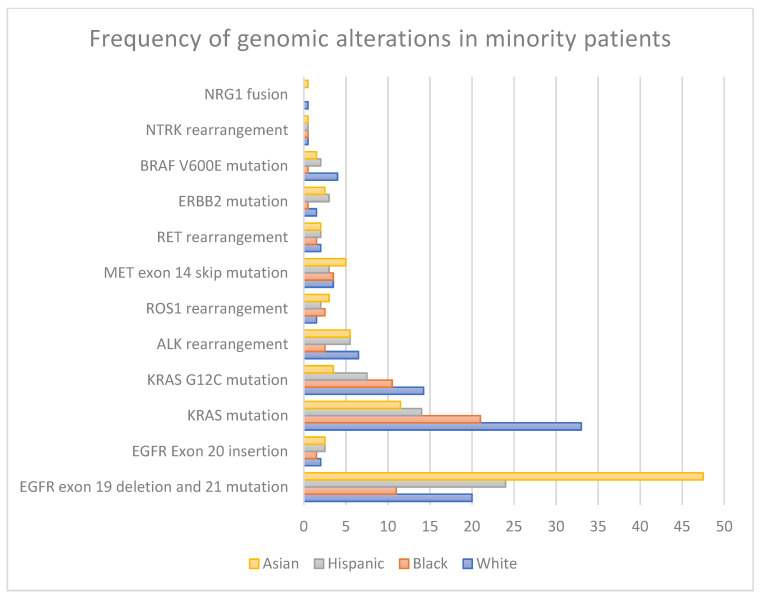
Genomic alteration prevalence across racial groups.

**Figure 2 cancers-17-01950-f002:**
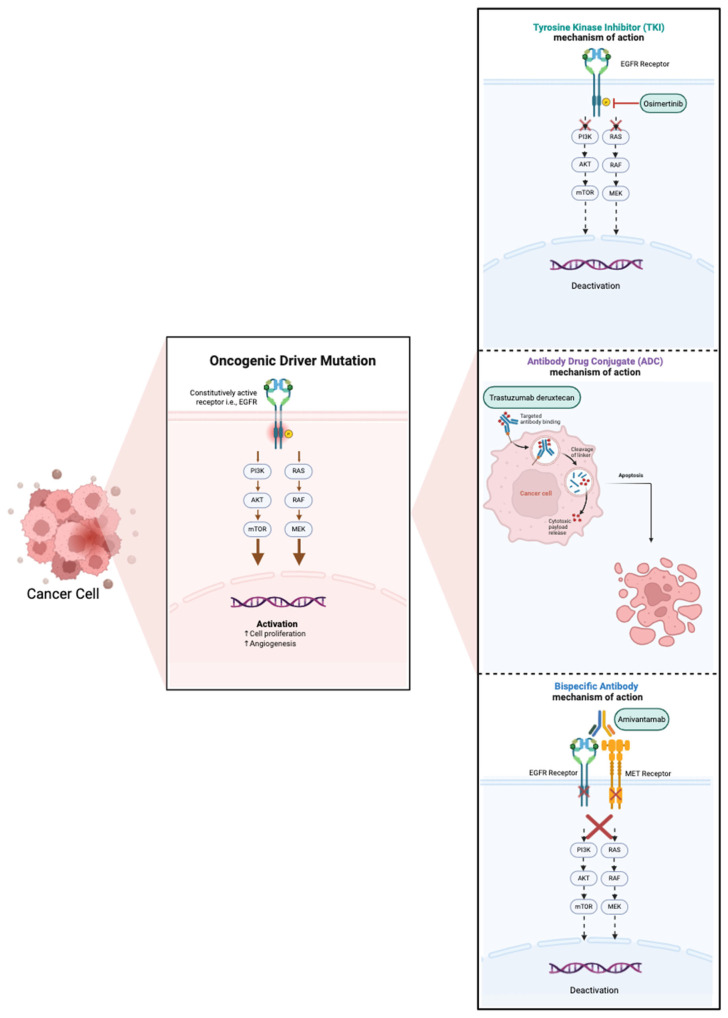
Mechanism of action of current FDA-approved targeted therapies.

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
