# Peer review of "Molecular Diagnosis, Clinical Trial Representation, and Precision Medicine in Minority Patients with Oncogene-Driven Lung Cancer"

_cancers, 2025, doi:10.3390/cancers17121950_

Round 1

Reviewer 1 Report

Comments and Suggestions for Authors

This manuscript describes NSCLC mutational status in minor populations in comparison with majority (white, caucasian, hispanic, asian populations) and participation of these in various clinical trials of targeted agents and outcomes of the trials.

Title of the manuscript is misleading and should be revised. should include keywords such as clinical trial and mutational status etc. 

subsection 2 - health disparities should be replaced with more appropriate title for this section

Ref. no. 71 - citation details are incomplete. 

In section 3 - contents of table 1,2, and 3 have been explained again. Table 1 - EGFR mutations status in various subpopulations is again explained in main text - line 171-176, 220-223. this is redundant and should be removed. 

Table 2 - It's mentioned in footnote that C stands for chemotherapy. But exact name of chemo drug isn't given except for docetaxel. Names of other chemo drugs should be included too.  

section 3.2 and 3.3 simply mention underrepresentation of minor populations in the clinical trials of the specific targeted agents and do not shed much light on treatment outcome in these populations. there is no mention of minor populations in clinical trials for other targeteable mutations such as ROS1 (line 314-328), HER2 mutations (line 356-364), NTRK rearrangements (line 378-382). 

as per list of abbreviations - NRTK stands for neurotrophic tyrosine receptor kinase, but at many places its wrongly mentioned as NTRK in the manuscript. (line 52, 378). Authors should use abbrev same throughout. 

section 5 appears to be an effort to address more of a social science issue. 

There is not a single figure in the review to make understanding for the audience more clear.  

Comments on the Quality of English Language

Minor English editing is required. e.g. line no. 420 - 'Molecularly' targeted therapy is not a correct word. 

Author Response

  1. Title of the manuscript is misleading and should be revised. should include keywords such as clinical trial and mutational status etc. 

Reply: Thank you for your feedback. We have revised the manuscript title as: “Molecular Diagnosis, Clinical Trial Representation and Precision Medicine in Minority Patients with Oncogene-Driven Lung Cancer”. This updated title includes references to both mutational status and clinical trials, aligning with the manuscript’s core themes. 

  1. Subsection 2 - health disparities should be replaced with more appropriate title for this section

Reply: Thank you for your feedback. The section title has been updated  to “Impact of racial and socioeconomic disparities on NSCLC Outcomes.”  (now Subsection 3 in the revised manuscript). 

  1. no. 71 - citation details are incomplete. 

Reply: Thank you for identifying this issue. The citation for reference 71 (now reference 65 in the revised manuscript) has been updated .

  1. In section 3 - contents of table 1,2, and 3 have been explained again. Table 1 - EGFR mutations status in various subpopulations is again explained in main text - line 171-176, 220-223. this is redundant and should be removed. 

Reply:  We appreciate your observation. To address redundancy, we have revised the text to reduce overlap with Tables 1–3 and streamlined descriptions to avoid repetition while preserving necessary context. 

  1. Table 2 - It's mentioned in footnote that C stands for chemotherapy. But exact name of chemo drug isn't given except for docetaxel. Names of other chemo drugs should be included too.  

Reply: Thank you for your comment. We have clarified the term “chemotherapy” in Table 2 to specify platinum-based chemotherapy. Where data was available (e.g., from real-world studies), we have listed the chemotherapy agents used, such as platinum agents, taxane agents, gemcitabine, and pemetrexed. 

  1. section 3.2 and 3.3 simply mention underrepresentation of minor populations in the clinical trials of the specific targeted agents and do not shed much light on treatment outcome in these populations. there is no mention of minor populations in clinical trials for other targetable mutations such as ROS1 (line 314-328), HER2 mutations (line 356-364), NTRK rearrangements (line 378-382). 

Reply: Thank you for your feedback. We acknowledge the limited data on treatment outcomes for minority populations with rare mutations like ROS1, HER2, and NTRK due to their low prevalence and challenges in recruiting sufficient patients for robust analyses. In the revised manuscript, we have expanded the discussion to explicitly note the scarcity of real-world and clinical trial data for these mutations in minority populations, which limits subgroup analysis. . 

  1. As per list of abbreviations - NRTK stands for neurotrophic tyrosine receptor kinase, but at many places its wrongly mentioned as NTRK in the manuscript. (line 52, 378). Authors should use abbrev same throughout. 

Reply: Thank you for identifying the incorrect abbreviation. We have corrected all instances of “NRTK” to the proper abbreviation “NTRK.”

  1. section 5 appears to be an effort to address more of a social science issue. 

Reply: Thank you for your feedback. To better align with the manuscript’s focus on precision oncology, we have restructured Section 5 (now split into Sections 3 and 4). Section 3 now addresses the impact of racial and socioeconomic disparities, integrating biological and social determinants, while Section 4 (“Future Directions”) focuses on actionable strategies to improve clinical trial diversity and equity in precision medicine, maintaining a clinical and translational focus.

  1. There is not a single figure in the review to make understanding for the audience more clear.

Reply: Thank you for your suggestion. We have added Figure 1, illustrating the frequency of oncogenic alterations by population, and Figure 2, detailing the mechanisms of FDA-approved targeted therapies

Reviewer 2 Report

Comments and Suggestions for Authors

Dear editor 

The manuscript entitled "Molecular Diagnosis and Precision Medicine in Minority Patients with Oncogene-Driven Lung Cancer" discuss about molecular diagnosis and treatment for minority patients with oncogene-driven lung cancer. This manuscript can be considered for publication after major revision and addressing following comments point by point:

1-The Manuscript is poorly prepared in term of figures. Please add a graphical abstract and some figures for each section considering copyright.

2- The conclusion is poorly written.Please complete it.

3- Please merge complete and modified version of "Future perspective" with conclusion.

4- The manuscript seems less critical. Pleas add a critical section separately , which explains and discusses about shortcomings or benefits of studies.

5- Add a paragraph explantation for section 3 (Treatment of minority patients....) before starting the subclassification 

6-It seems mutations classifications is so incomplete. Some of them are mentioned in 3.4 section which must be considered as a separate secion. Please Add more studies.

Comments on the Quality of English Language

The english should be polished as well.

Author Response

  1. The Manuscript is poorly prepared in term of figures. Please add a graphical abstract and some figures for each section considering copyright.

Reply: Thank you for your suggestion. We have added a graphical abstract summarizing the key themes of the review and included two figures to visually enhance the manuscript: Figure 1: Genomic alteration frequencies by racial/ethnic group; Figure 2: Mechanisms of current FDA-approved targeted therapies . 

  1. The conclusion is poorly written. Please complete it. Please merge complete and modified version of "Future perspective" with conclusion.

Reply: Thank you for your feedback. We have revised the conclusion to integrate core points from the former “Future Perspective” section, and also created a standalone Discussion section to provide a cohesive summary and critical insight.

  1. The manuscript seems less critical. Pleas add a critical section separately, which explains and discusses about shortcomings or benefits of studies.

Reply: Thank you for your suggestion. We have added a Discussion section to provide critical analysis on clinical trial data, real-world outcomes, and gaps in current knowledge - especially pertaining to minority populations. This section highlights the benefits of targeted therapies but critiques the underrepresentation of minority patients, limited racial subgroup analyses, and challenges in studying rare mutations due to small sample sizes. It also discusses socioeconomic barriers affecting trial access and testing disparities, providing a balanced perspective on study limitations and clinical advancements .

  1. Add a paragraph explanation for section 3 (Treatment of minority patients....) before starting the subclassification 

Reply: Thank you for your suggestion. A new introductory paragraph has been added at the beginning of Section 2 (formerly Section 3) to set the stage for the clinical trial and treatment discussions. 

  1. It seems mutations classifications is so incomplete. Some of them are mentioned in 3.4 section which must be considered as a separate section. Please Add more studies.

Reply: Thank you for your feedback. We have expanded Section 2.4 and created a clearer structure for rare mutations, including NRG1, NTRK, ERBB2, and BRAF, with updated real-world and trial data, where available.

Reviewer 3 Report

Comments and Suggestions for Authors

The review is comprehensive and appropriately referenced. 

1) However, there is an error in Table 1.  Under the line "Exon 20 Insertion" the percentages listed are the percentages of EGFR insertion 20 mutations among all EGFR mutations.  For example, the "9%" listed for Caucasians is the actually 9% of all EGFR mutations.  This turns out to be about 2% of all cases as actually stated in reference 27.  The values across the table for all racial groups in this category are also incorrect.

2) There needs to be some comments on NRG1 fusions as these are targetable by FDA approved agents for NSCLC. These may be uncommon enough to make racial categorization difficult, but even that should be acknowledged in a comprehensive review.

Author Response

  1. However, there is an error in Table 1.  Under the line "Exon 20 Insertion" the percentages listed are the percentages of EGFR insertion 20 mutations among all EGFR mutations.  For example, the "9%" listed for Caucasians is the actually 9% of all EGFR mutations.  This turns out to be about 2% of all cases as actually stated in reference 27.  The values across the table for all racial groups in this category are also incorrect.

Reply: Thank you for identifying the discrepancy. We have corrected the values to represent prevalence among all NSCLC patients, not just within EGFR mutations, and have verified the figures against original sources.

  1. There needs to be some comments on NRG1 fusions as these are targetable by FDA approved agents for NSCLC. These may be uncommon enough to make racial categorization difficult, but even that should be acknowledged in a comprehensive review.

Reply: Thank you for your suggestion. We have added a discussion of NRG1 fusions in Table 1 and Section 2, noting their rarity and the recent FDA approval of zenocutuzumab for NRG1 fusion-positive NSCLC.

Reviewer 4 Report

Comments and Suggestions for Authors

The article is well-written and provides valuable insights for readers. I would like to suggest that the authors consider including two figures: (1) a molecular mechanism illustrating how the described mutation contributes to drug resistance, and (2) a schematic representation of potential molecular targets associated with this mutation.

Beside this there are many artcile also suggest that Non coding RNA is a good biomarker for the lung cancer Non-Coding RNA 202410(5), 50; https://doi.org/10.3390/ncrna10050050,  doi: 10.18632/aging.103496, https://doi.org/10.1016/j.biopha.2021.112190

I would recommend including one section for the noncoding RNA with a Table and writing that will enhance the manuscript quality

These additions would enhance the clarity and impact of the manuscript, particularly for readers interested in translational applications.

Author Response

  1. I would like to suggest that the authors consider including two figures: (1) a molecular mechanism illustrating how the described mutation contributes to drug resistance, and (2) a schematic representation of potential molecular targets associated with this mutation.

Reply: Thank you for your suggestion. We have included: Figure 1: Genomic alteration prevalence across racial groups; Figure 2: Mechanisms of Action of targeted therapies, including resistance pathways and molecular targets.

  1. Beside this there are many article also suggest that Non coding RNA is a good biomarker for the lung cancer Non-Coding RNA2024, 10(5), 50; https://doi.org/10.3390/ncrna10050050,  doi: 10.18632/aging.103496, https://doi.org/10.1016/j.biopha.2021.112190

I would recommend including one section for the noncoding RNA with a Table and writing that will enhance the manuscript quality

Reply: Thank you for your suggestion. We have added a brief subsection under “Molecular Diagnosis” discussing non-coding RNAs as emerging biomarkers, with relevant references.

Round 2

Reviewer 1 Report

Comments and Suggestions for Authors

None

Reviewer 2 Report

Comments and Suggestions for Authors

The manuscript revised as well and can be published in present form.

Reviewer 4 Report

Comments and Suggestions for Authors

Author improved manuscript 

Comments on the Quality of English Language

need polishing